# A ligand-directed divergent catalytic approach to establish structural and functional scaffold diversity

Yen-Chun Lee[1,2], Sumersing Patil[1,2], Christopher Golz[2], Carsten Strohmann[2], Slava Ziegler[1], Kamal Kumar[1] & Herbert Waldmann[1,2]

The selective transformation of different starting materials by different metal catalysts under individually optimized reaction conditions to structurally different intermediates and products is a powerful approach to generate diverse molecular scaffolds. In a more unified albeit synthetically challenging strategy, common starting materials would be exposed to a common metal catalysis, leading to a common intermediate and giving rise to different scaffolds by tuning the reactivity of the metal catalyst through different ligands. Herein we present a ligand-directed synthesis approach for the gold(I)-catalysed cycloisomerization of oxindole-derived 1,6-enynes that affords distinct molecular scaffolds following different catalytic reaction pathways. Varying electronic properties and the steric demand of the gold(I) ligands steers the fate of a common intermediary gold carbene to selectively form spirooxindoles, quinolones or *df*-oxindoles. Investigation of a synthesized compound collection in cell-based assays delivers structurally novel, selective modulators of the Hedgehog and Wnt signalling pathways, autophagy and of cellular proliferation.

[1] Max-Planck-Institut für Molekulare Physiologie, Abteilung Chemische Biologie, Otto-Hahn-Straße 11, Dortmund 44227, Germany. [2] Technische Universität Dortmund, Fakultät Chemie, Chemische Biologie, Otto-Hahn-Straße 6, Dortmund 44221, Germany. Correspondence and requests for materials should be addressed to K.K. (email: kamal.kumar@mpi-dortmund.mpg.de) or to H.W. (email: herbert.waldmann@mpi-dortmund.mpg.de).

In the design and assembly of compound collections for chemical biology and medicinal chemistry research, the structural diversity of the core molecular scaffolds of the library members[1,2] is decisive to guarantee high performance in biological investigations and to deliver structurally and functionally diverse small molecules[3]. Therefore, novel methods for the efficient synthesis of structurally distinct and diverse 'privileged' molecular scaffolds giving rise to compound classes with differing and selective bioactivity are in very high demand[4–6], yet remain among the most challenging problems putting organic synthesis itself at the heart of discovery in chemical biology and medicinal research[7–9].

In established approaches for the synthesis of structurally different molecular scaffolds, usually starting materials are varied in structure and subjected to transformation by different reagents and catalysts, in particular metal catalysts under individually established reaction conditions. Depending on the structure of the individual starting material, the reaction conditions, the reagents and catalysts employed, structurally different intermediates and products may be formed[10–12]. In this approach, chemical diversity is established individually and step-wise via structural modification of the starting materials, the reagents and the catalyst in series of isolated experiments, which renders it laborious and time-consuming. An alternative and more unified approach would utilize common starting materials that when exposed to a common mode of metal catalysis lead to a common type of intermediate, whose molecular fate then is steered to yield structurally different scaffolds by tuning the reactivity of the metal catalyst through different ligands. We demonstrate that this unified ligand-directed divergent synthesis approach can provide facile and efficient access to structurally rich compound collections that in turn can provide small molecules with different and selective biological activity. Key to this approach is the identification of both efficient and flexible transformations that preferably proceed through one of the several possible intermediates of a given catalytic cycle, whose molecular fate can be directed into different reaction pathways leading to different molecular frameworks. We report that use of different ligands in gold(I)-catalysed transformations of oxindole-derived 1,6-enynes (1) steers the common gold carbene intermediate formed by 6-*endo*-dig cyclization to the selective formation of three distinct natural product-inspired scaffolds and via (Fig. 1) biological investigation of resulting structurally rich compound collection in cell-based assays reveals structurally novel, selective modulators of the Hedgehog (Hh) and Wnt signalling pathways, autophagy and of cellular proliferation.

## Results

**Design and optimization of catalytic divergent synthesis.** Gold(I)-mediated cycloisomerization reactions of 1,n-enynes (n = 5–7) are efficient molecular rearrangements that often pass through cyclopropyl gold carbene intermediates[13–16], whose reactivity can be influenced by the properties of the ligands in the gold complex and by their substitution pattern[17–19]. Therefore, a wealth of gold and other coinage metal-catalysed transformations to generate distinct molecular scaffolds has been reported in recent years[20–29]. For gold(I)-catalysed transformations numerous different ligands, such as phosphines, phosphites, heterocyclic carbenes, halides and others are readily available. To explore whether the cyclopropyl gold carbene intermediates[30–32] formed in the ene-yne cyclization could be directed into diverse reaction pathways by modulating the ligand structure and at the same time arriving at biologically relevant scaffolds[33,34], skeletal rearrangements of oxindole-derived 1,6-enynes (Fig. 1) were investigated. The oxindole scaffold is the structurally determining entity of a large class of natural products endowed with numerous bioactivities and has served as blueprint enabling the discovery of novel biologically active natural product-inspired compound classes[35–38].

To explore the feasibility of the approach, oxindole-derived 1,6-enyne (E:Z = 3:1) **1a** was prepared from isatin by the addition of lithium phenylacetylide to the keto group and subsequent O-allylation of the resulting tertiary alcohol (Supplementary Methods). Treatment of the 1,6-enyne **1a** with 5 mol% of the cationic gold(I) complexes Au(OTf)PPh$_3$ or Au(BF$_4$)PPh$_3$ in dichloromethane (DCM) yielded spirooxindole **2a** in low to moderate yield (entries 1–2, Table 1). In contrast, exposure of the enyne to AuCl$_3$ under the same conditions led to the formation of structurally distinct cyclopropyl-substituted quinolone **3a** in low yield (entry 3). Employment of a gold complex with an N-heterocyclic carbene bearing a highly electron donating ligand (**I**, entry 4), gave a mixture of **2a** and **3a**. Gratifyingly, treatment with the sterically demanding gold complexes **II–IV** in DCM selectively delivered **3a** (entries 5–7, Table 1). Catalyst **III** yielded *syn*-quinolone **3a** as major product in a preparatively viable yield of 67% (entry 6, Table 1). The bulk of the biaryl moiety seems critical for the efficiency of the reaction since gold complexes **II** and **IV** embodying an unsubstituted biaryl ring (R$^1$ = H), but sterically equally or more demanding phosphines gave lower yields than complex **III** (entry 5,7).

We reasoned that employment of a more electrophilic gold complex like **V**, employed by Nieto-Oberhuber *et al.* in [4 + 2] cycloaddition reactions of 1,6-enynes[18] would favour migration

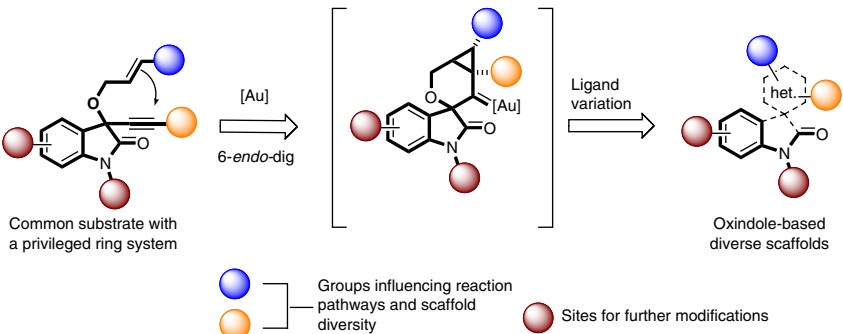

**Figure 1 | Design of ligand-directed gold catalysis approach to diverse scaffolds.** The approach explores the potential of a common substrate, supporting a privileged oxindole ring system and carrying enyne functional groups that can be influenced differently to follow distinct reaction pathways and thereby generating distinct scaffolds. In particular, ligand variations in the catalytic gold complex can steer the common spirooxindole gold carbene intermediate, formed by gold(I)-mediated 6-*endo*-dig cycloisomerization, to different rearrangements and leading to diverse scaffolds. Het, heterocycle.

**Table 1 | Ligand-directed distinct cycloisomerizations of *O*-crotylated substrate 1a to yield different scaffolds.**

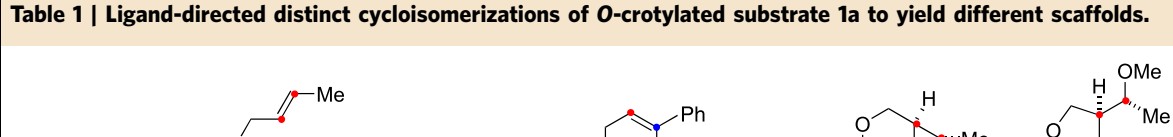

| Entry | [Au] | Solvent | Nu. (eq) | Temp. | Yield (%)* | | |
|---|---|---|---|---|---|---|---|
| | | | | | **2a** | **3a** | **4a** |
| 1 | Au(OTf)PPh₃ | DCM | — | rt | 20 | — | — |
| 2 | Au(BF₄)PPh₃ | DCM | — | rt | 33 | — | — |
| 3† | AuCl₃ | DCM | — | rt | — | 23 | — |
| 4 | **I** | DCM | — | rt | 43 | 7 | — |
| 5 | **II** | DCM | — | rt | — | 57 | — |
| 6 | **III** | **DCM** | — | **rt** | — | **67** | — |
| 7 | **IV** | DCM | — | rt | — | 43 | — |
| 8 | **V** | **DCM** | — | **rt** | 60 | — | — |
| 9 | **V** | DCE | MeOH (20.0) | 60 °C | | | 20 |
| 10 | **II** | DCE | MeOH (20.0) | 60 °C | — | — | 62 |
| 11 | **III** | DCE | MeOH (20.0) | 60 °C | — | — | 56 |
| 12 | **IV** | DCE | MeOH (20.0) | 60 °C | — | — | 50 |
| 13 | **II** | **DCE** | **MeOH (10.0)** | **60 °C** | — | — | **73** |

DCE, dichloroethane; eq, equivalent; Mes, mesityl; Nu., nucleophile; rt, room temperature; temp., temperature. Bold text indicates best optimized reaction conditions.
*Isolated yields.
†Starting material recovered.

of the cyclopropane ring conserved in **3a**. Gratifyingly, 5 mol% of gold complex **V** at room temperature in DCM led to the selective formation of spirooxindole **2a** in 60% yield (entry 8, Table 1).

To induce additional rearrangements of a plausible cyclopropane gold carbene intermediate, enyne **1a** was exposed to catalysts **II–V** in the presence of methanol as nucleophile that might trigger cyclopropane ring opening. Indeed, treatment with catalyst **V** in dichloroethane (DCE) at 60 °C yielded a novel product embodying the (*E*)-3-(dihydrofuran-2(3*H*)-ylidene) indolin-2-one scaffold (*df*-oxindole, entry 9) in low yield. Again, using the related gold complexes **II–IV** under the same reaction conditions revealed that a right balance of steric bias and electrophilic nature is the key to selectively form the *df*-oxindoles (entries 9–12). Thus, catalyst **II** with bulky groups on the phosphine and unsubstituted biphenyl moiety afforded the *df*-oxindole **4a** in viable yield (entry 10), while catalysts **III** and **IV** with sterically demanding groups on the biphenyl moiety and the phosphine were less efficient (entries 11–12). Finally, reducing the amount of methanol gave the best yield of *df*-oxindole **4a** (73%, entry 13, and Supplementary Table 1). The structures of products **2a–4a** with different scaffolds were unambiguously ascertained by X-ray crystal structure analysis (Supplementary Tables 3–5).

These results demonstrate that variation of the ligand under otherwise nearly identical reaction conditions does allow to steer the fate of a most likely common cyclopropyl gold carbene

intermediate (see below for a mechanistic rationale) into different trajectories giving rise to distinct and structurally diverse molecular scaffolds. The three scaffolds formed from the 1,6-enynes represent characteristic structural elements of different drug and natural product classes, and thus cover different areas of biologically relevant chemical space, raising the expectation that they may also yield novel modulators of different biological phenomena[39–43].

**Construction of a scaffold-rich compound collection.** Variation of the oxindole substituents on the aryl ring, at the nitrogen and on the acetylene moiety allowed to rapidly synthesizing a small, but structurally diverse compound collection (Fig. 2). In all cases, the products were formed stereospecifically and in good to acceptable yields. In general, variation of electron density in the aryl ring of the oxindole and on the acetylene was well tolerated. 1,6-enynes with a thiophene as a representative heterocycle attached to the acetylene were also efficiently transformed into three different scaffolds (**2e**, **3c** and **4e**, Fig. 2).

**Mechanistic insights into gold-catalysed transformations.** By analogy to previous observations for related gold-catalysed cyclorearrangements[44–46], we assume that formation of scaffolds **2–4** plausibly occurs via a common spirooxindole gold carbene **6** formed by 6-*endo*-dig cyclization of enynes **1** (Fig. 3). Of the three

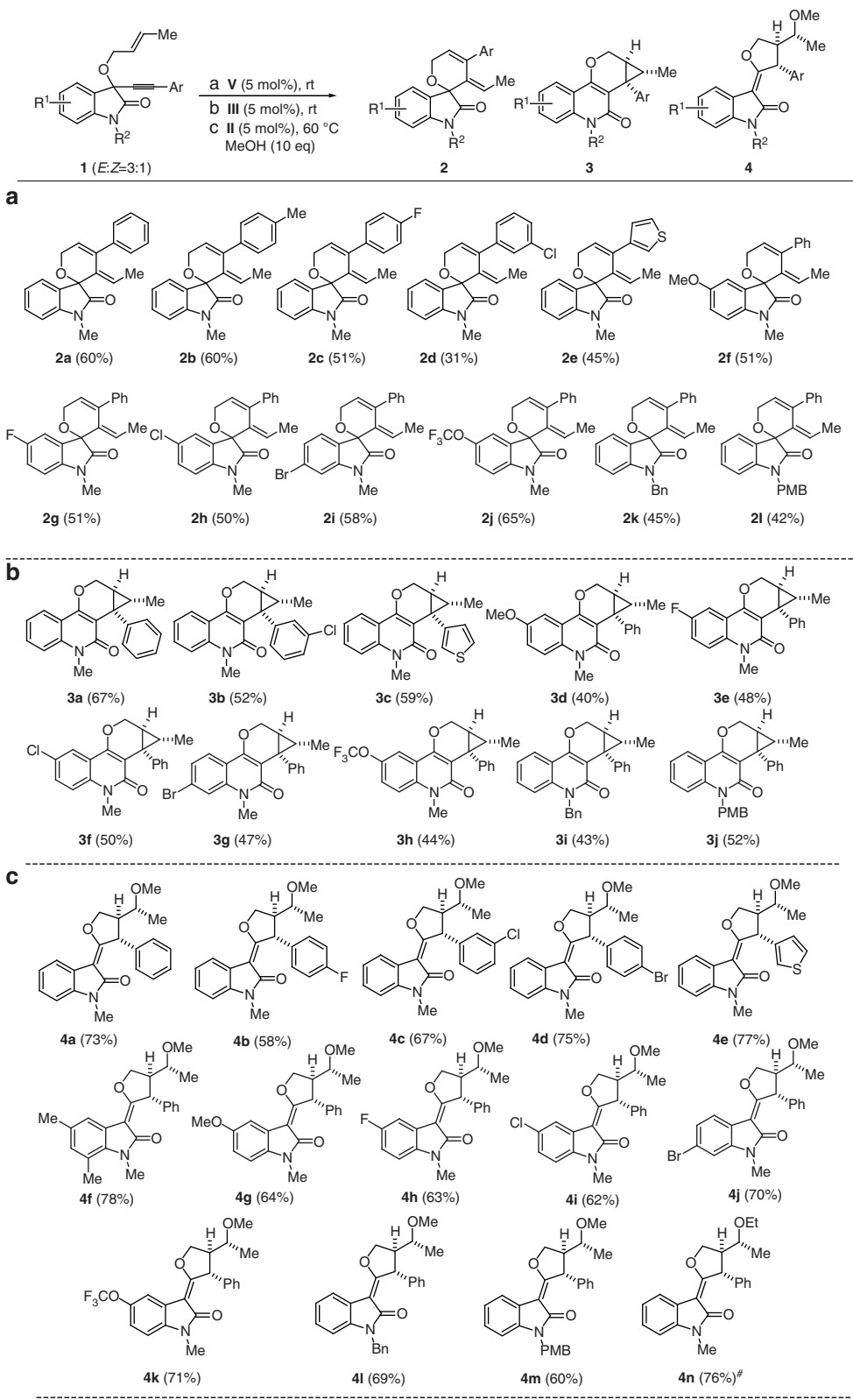

**Figure 2 | A collection of diverse small molecules obtained via LDS appraoch.** (**a**) Synthesis of spirooxindoles **2**: 1,6-enynes **1** (0.1 M), **V** (0.5 mol%), DCM and 0 °C to rt, overnight. (**b**) Synthesis of quinolones **3**: 1,6-enynes **1** (0.1 M), **III** (0.5 mol%) and DCM, 0 °C to rt, overnight. (**c**) Synthesis of *df*-oxindoles **4**: 1,6-enynes **1** (0.1 M), MeOH (10 eq), **II** (0.5 mol%), DCE and rt to 60 °C, overnight. Bn, benzyl; DCE, dichloroethane; eq, equivalent; LDS, ligand-directed divergent synthesis; PMB, *p*-phenoxy benzyl; rt, room temperature.

**Figure 3 | Proposed mechanistic pathways of divergent gold(I) catalysis to afford different scaffolds.** (**a**) The spirooxindole gold carbene intermediate **6** was formed by the cationic gold(I) activation of alkyne **1**, followed by 6-*endo*-dig cyclization and cyclopropane formation. The most electrophilic gold(I) catalyst **V** induces a sequential cyclopropane migration and deauration to generate spirooxindole **2** (magenta arrows). Very bulky gold complex **III** promotes the pinacol type acyl migration followed by deauration to afford quinolone **3** (blue arrows). In the presence of MeOH, the less bulky but relatively more electrophilic gold complex **II** allows the 1,4-nucleophilic addition of methanol to open up the cyclopropane ring (**10**). Subsequent protonation, spiroether insertion and deauration in **10** generate the *df*-oxindole **4** (turquoise arrows). (**b**) A comparison of the molecular properties of the cationic gold catalysts **II**, **III** and **V**.

catalysts yielding the products with different scaffolds with highest yields, gold complex **V** is most electrophilic. In the presence of this catalyst, cyclopropane ring migration may be favoured to yield rearrangement product **7** or the more stable benzylic carbocation **8** that ultimately will lead to the spirooxindole **2** (magenta arrows, Fig. 3). We furthermore assume that in the presence of the very bulky gold complex **III**, intermediate **6** tends to relieve the strain of the spiro-ring by means of a pinacol type acyl migration thereby forming oxonium–quinolone gold intermediate **9**. Subsequent deauration leads to the formation of tetracyclic quinolone **3** (blue arrows, Fig. 3). In the presence of the relatively less bulky but more electrophilic gold complex **II**, intermediate **6** will be relatively more stabilized to favour nucleophilic addition with cyclopropane ring opening (**10**) followed by protonation to yield *anti*-gold carbene intermediate **11** (see also **18** in Fig. 5a). Intramolecular nucleophilic attack by the spiroether will lead to highly strained and reactive tetracyclic oxonium ion **12** that by deauration and concomitant ring opening will give rise to *df*-oxindole **4** (turquoise arrows, Fig. 3)[47].

In all rearrangements of 1,6-enynes **1** (Fig. 3), the substitution pattern on the olefin will have a major influence on the stability of carbocation intermediate **5** and susceptibility of the cyclopropane ring in intermediate **6** to nucleophilic attack. Thus, we reasoned that increasing the stability of the intermediary carbocation might open paths to different rearrangements and scaffolds. To explore this notion, dimethyl substituted enynes **13** were exposed to different gold complexes, and gratifyingly treatment with highly electrophilic gold complex **II** in DCM selectively yielded structurally different *df*-oxindoles **14**. Solvent variation (Supplementary Table 2) revealed that in tetrahydrofuran product **14a** was formed in almost quantitative yield as a single diastereoisomer (Fig. 4).

**Synthesis of *df*-oxindoles from *O*-prenylated substrates**. Formation of *df*-oxindoles **14** occurs by an analogous initial 6-*endo*-dig cyclization of the olefin to the acetylene activated by gold complex **II** to form cyclopropane gold carbene **15** (see Fig. 5a for conversion of **13** into **14**). However the *gem*-dimethyl substitution facilitates cyclopropane ring opening via relatively stable carbocation **16** and proton loss to yield intermediate **17**. Protonation yields *anti*-gold carbene intermediate **18** that sets the stage for the formation of analogous oxiranium ion **19** and final rearrangement to unsaturated *df*-oxindoles **14** (Fig. 5). This mechanistic proposal is supported by deuterium incorporation and formation of the corresponding deutero-methyl ether in the presence of CD$_3$OD (Fig. 5b). The structure of *df*-oxindole **14a** was unambiguously ascertained by X-ray crystal structure analysis (Supplementary Table 6).

The rearrangement tolerates different substituents on the indole and the acetylenic moiety, including thiophene as representative heterocycle (**14g**) and affords products **14** as single *anti*-diastereoisomers (Fig. 4). Alkyl-substituted acetylenes exhibit slower conversion, but in the presence of 10 mol% catalyst the corresponding *df*-oxindoles (**14h**–**i**) were formed stereoselectively and in good yields. By analogy variation of the substituents on the aryl part of the oxindole and of the N-protecting group yielded the corresponding *df*-oxindoles (**14j**–**p**) and (**14q**–**u**), respectively, in good to excellent yields as single diastereoisomers.

**Discovery of small molecules with orthogonal biological activities**. To explore if the scaffold diversity in the compound collection translates into selective biological activity, the synthesized compound collection was exposed to cellular assays monitoring different biological phenomena. Much to our delight we identified individual compounds that are structurally novel

**Figure 4 | Scope of the gold catalysed rearrangement of *O*-prenylated substrate 13 to *df*-oxindoles 14.** Reaction condition: 1,6-enynes **13** (0.1 M), **II** (0.5 mol%), THF, 0 °C to rt, overnight. MOM, methoxymethyl; SEM, (2-(trimethylsilyl)ethoxy)methyl; THF, tetrahydrofuran.

selective inhibitors of the Hh and Wnt signalling pathways, autophagy and of cellular proliferation and displayed largely orthogonal biological activity.

The Hh pathway is an evolutionarily conserved developmental signalling pathway that governs vital processes such as cell proliferation, differentiation, body patterning, tissue repair and regeneration[48]. In the absence of ligand, the membrane receptor Patched represses the seven-transmembrane protein Smoothened (Smo). The Hh pathway is activated on binding of the Hh ligand to Patched-1, thereby relieving the inhibition of Smo. Through a series of events that occur in the primary cilium, Smo promotes activation of Gli transcription factors that drives the transcription of Hh target genes. Aberrations in Hh signalling are associated with birth defects and cancer, including medulloblastoma and basal cell carcinoma[48]. Therefore, development of small-molecule modulators of Hh pathway is of utmost importance in cancer research.

Hh pathway inhibitors were identified by means of an osteogenesis assay that monitors Hh signalling activity in pluripotent mouse mesenchymal C3H10T1/2 cells on stimulation with purmorphamine[49]. Several oxindoles inhibited osteogenesis with half-maximal inhibitory concentrations ($IC_{50}$) in the low micromolar range, as detected by reduced activity of the osteogenic marker alkaline phosphatase.

The most potent compounds **14q** ($IC_{50} = 2.8 \mu M$) and **14r** ($IC_{50} = 3.1 \mu M$; Fig. 6a; Supplementary Table 7) were additionally characterized in the orthogonal Gli reporter gene assay using Shh

Light II cells, a NIH/3T3-derived cell line that is stably transfected with Gli-responsive firefly luciferase and constitutive Renilla luciferase reporters[50]. **14q** and **14r** inhibited the Gli-dependent luciferase activity with $IC_{50}$ values of 1.7 and 0.8 μM, respectively (Fig. 6b). Both compounds did not affect the viability of C3H10T1/2 cells and Renilla luciferase activity in Shh Light II cells, therefore, suggesting a specific inhibition of Hh signalling.

To further confirm and characterize Hh pathway inhibition, we monitored the expression of the Hh target genes *Ptch1* and *Gli1* on compound treatment[48]. Ptch1 is a negative regulator and Gli1, a positive regulator of Hh signalling, and both control the feedback regulation of the pathway. On treatment of C3H10T1/2 cells with the compounds together with purmorphamine the expression of *Ptch1* and *Gli1* was suppressed in a dose-dependent manner (Fig. 6c,d).

Several small molecules like vismodegib and cyclopamine that modulate Hh signalling target the seven-transmembrane receptor Smo[48]. Thus, we assessed the ability of **14q** and **14r** to bind to Smo by means of competition experiments employing boron-dipyrromethene (BODIPY)-cyclopamine in HEK293T cells ectopically expressing Smo[50]. **14q** and **14r** could displace BODIPY-cyclopamine as observed by the decrease in BODIPY fluorescence (Fig. 7a). Concentration-dependent displacement of BODIPY-cyclopamine by **14q** and **14r** was detected using flow cytometry (Fig. 7b), suggesting that **14q** and **14r** most likely inhibit the Hh pathway by directly binding to Smo.

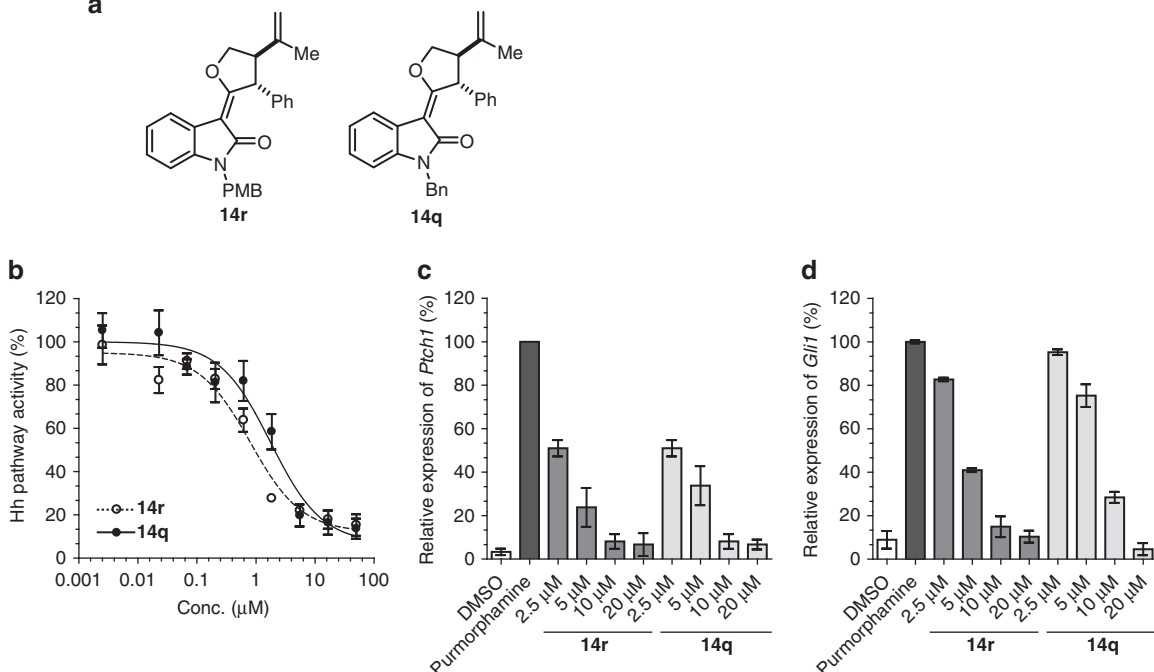

**Figure 5 | Mechanistic proposal for the formation of *df*-oxindole 14.** (**a**) The proposed mechanism of the formation of *df*-oxindole **14** involves the gold(I)-mediated 6-*endo*-dig cycloisomerization of 1,6-enyne to afford **15**, followed by subsequent deprotonation, protonation, formation of a strained tetracyclic spiro-oxenium intermediate (**19**) and its rearrangement to afford **14**. (**b**) In the presence of *d*-MeOH (as nucleophile as well as deuterium source), *df*-oxindole **14** and *d*-MeOH adduct **20** of *df*-oxindole was obtained in 14% yield with 40% deuteration at benzylic position and 47% yield with 50% deuteration at benzylic position, respectively. MeOH-*d*₄, deuterated methanol.

**Figure 6 | 14q and 14r inhibit Hh signalling.** (**a**) Structures of **14r** and **14q**. (**b**) **14r** and **14q** inhibit Gli-dependent reporter gene expression in Shh Light II cells. Cells were treated with 1.5 μM purmorphamine and different concentrations of compounds for 48 h. Firefly and Renilla luciferase activities were then determined and ratios of firefly luciferase/Renilla luciferase signals were calculated, which are a measure of Hh pathway activity. Nonlinear regression analysis was performed using a four parameter fit. **14r** and **14q** suppress the expression of the Hh target genes *Ptch1* (**c**) and *Gli1* (**d**). C3H10T1/2 cells were treated with purmorphamine (1.5 μM) and different concentrations of **14r** and **14q** or DMSO as a control for 48 h before isolation of total RNA. Following complementary DNA (cDNA) preparation, the relative expression levels of *Ptch1*, *Gli1* and *Gapdh* were determined by means of quantitative PCR employing specific oligonucleotides for *Ptch1* and *Gli1* or *Gapdh* as a reference gene. Expression levels of *Ptch1* and *Gli1* were normalized to the levels of *Gapdh* and are depicted as percentage of gene expression in cells activated with purmorphamine (100%). All data are mean values of three independent experiments (*n* = 3) ± s.d.

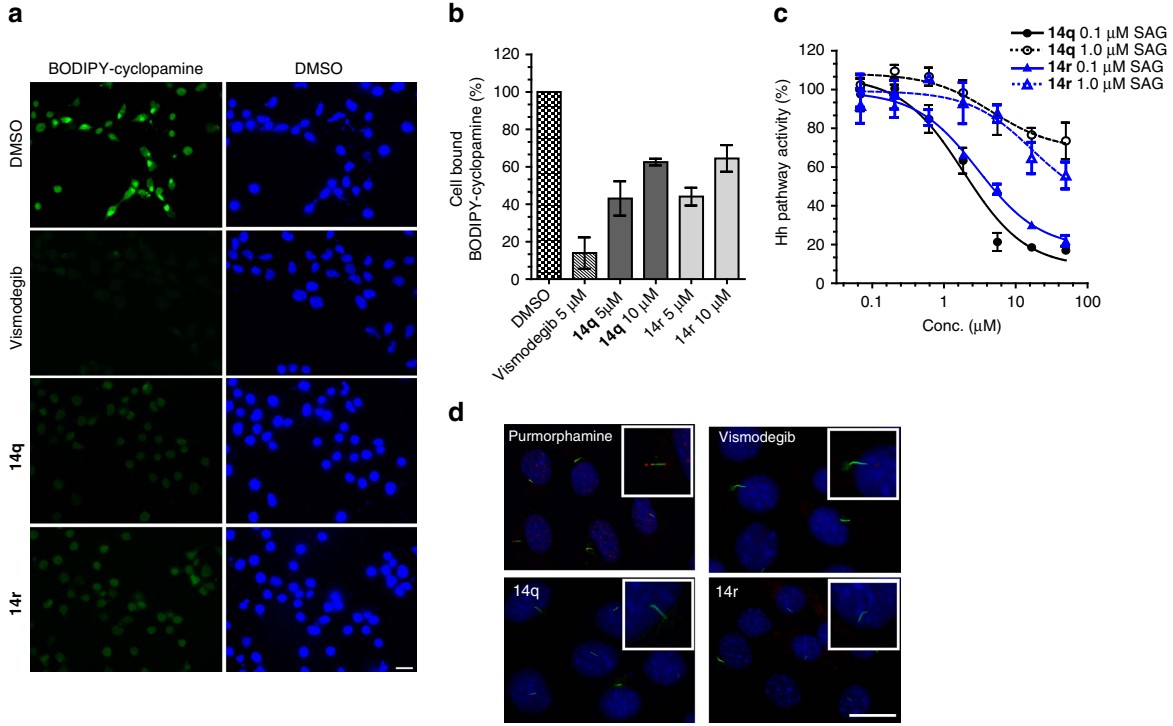

**Figure 7 | 14q and 14r inhibit Smo.** (**a**) **14q** and **14r** displace BODIPY-cyclopamine from Smo. HEK293T cells were transiently transfected with Smo expressing plasmid or empty vector. Forty-eight hours later cells were treated with BODIPY-cyclopamine (5 nM, green) followed by addition of 10 μM of **14q** or **14r** or vismodegib (5 μM) and DMSO as controls. Cells were incubated for 1 h before fixation and staining with DAPI (4′,6-diamidino-2-phenylindole) to visualize the nuclei (blue). Scale bar, 20 μm. (**b**) HEK 293T cells ectopically expressing Smo were treated with different concentration of the compounds (**14q** and **14r**), vismodegib or DMSO as controls in the presence of BODIPY-cyclopamine (5 nM) for 5 h. The graph shows the percentage of cell-bound BODIPY-cyclopamine as detected by fluorescence-activated cell sorting analysis. Data are mean values of three independent experiments ($n = 3$) ± s.d. (**c**) Influence of **14q** and **14r** on the Gli-mediated reporter gene expression on Hh pathway activation in Shh Light II cells by SAG (0.1 and 1 μM). Nonlinear regression analysis was performed using a four parameter fit. Data are mean values of three independent experiments ($n = 3$) ± s.d. and were normalized to cells treated with the respective concentration of SAG (set to 100%). (**d**) NIH/3T3 cells were treated with purmorphamine (1.5 μM) for 2 h followed by addition of vismodegib (2 μM), **14q** (5 μM) and **14r** (5 μM), and further incubation for 12 h. Cells were then fixed and stained to visualize the nuclei (DAPI, 4′,6-diamidino-2-phenylindole; blue), Smo (red) and cilia (acetylated tubulin; green). Inset: representative single cilia. Scale bar, 10 μm.

To further characterize inhibition of the Hh pathway by **14q** and **14r**, we activated the pathway with the Smo agonist SAG. High concentrations of SAG saturate the binding sites on Smo, resulting in decreased inhibitory activity of Smo antagonists (for example, vismodegib), whereas the action of Hh pathway inhibitors that act downstream of Smo (for example, GANT61) is not impaired (Supplementary Fig. 1)[51]. Indeed, the ability of **14q** and **14r** to inhibit the Gli-responsive reporter gene expression was reduced, when 1 μM SAG was used instead of 0.1 μM SAG (Fig. 7c), providing further proof that **14q** and **14r** inhibit Hh signalling by binding to Smo.

In vertebrate Hh signalling, Patched-1-induced inhibition of Smo is relieved on binding of the Hh ligand to Patched-1. Most of the Hh signalling cascade events occur in the primary cilium and involve translocation of multiple pathway components into this sensory organelle[48]. On Hh pathway activation Smo is laterally trafficked to the primary cilium via intraflagellar transport by binding to the kinesin-like protein KIF3a. However, genetic studies demonstrated that Smo is constantly cycling through the cilium without activating the pathway. Thus, localization of Smo into the cilium might be necessary, but not sufficient for Hh pathway activity[52]. To investigate the influence of the compounds on Smo trafficking to the primary cilium, we visualized Smo and acetylated tubulin as a ciliary marker. Purmorphamine promotes the localization of Smo in the primary cilium (Fig. 7d). However, on treatment with **14q** and **14r**, the purmorphamine-induced

ciliary localization of Smo is prevented similarly to the treatment with the Smo antagonist vismodegib (Fig. 7d).

To rule out any covalent binding of *df*-oxindoles causing inhibition of Hh signalling, a wash out step was included 30 min post addition of the compounds in the Gli reporter gene assay and the BODIPY-cyclopamine displacement experiment. Removal of the compounds almost completely abolished their activity to suppress the expression of the reporter gene (Supplementary Fig. 2a) and to displace BODIPY-cyclopamine (Supplementary Fig. 2b). Thus, most likely **14q** and **14r** reversibly modulated Hh signalling.

As example for a different major cellular signalling cascade, we analysed whether the compound collection contains selective inhibitors of the Wnt signalling pathway. Wnt signalling is involved in the regulation of cell proliferation, migration and polarity, tissue regeneration and stem cell renewal, and is a major pathway with relevance to the establishment of cancer[53]. Wnt signalling modulators are widely employed to dissect signal progression through the pathway[53]. In canonical Wnt signalling, the absence of a Wnt signal results in a low protein level of the central player β-catenin. In the nucleus, transcription factors of the T-cell factor/lymphoid enhancer factor (TCF/LEF) family inhibit transcription of Wnt target genes by recruiting histone deacetylases. On Wnt activation, β-catenin accumulates in the cytoplasm and translocates to the nucleus, where it associates with TCF/LEF and recruits transcriptional coactivators and

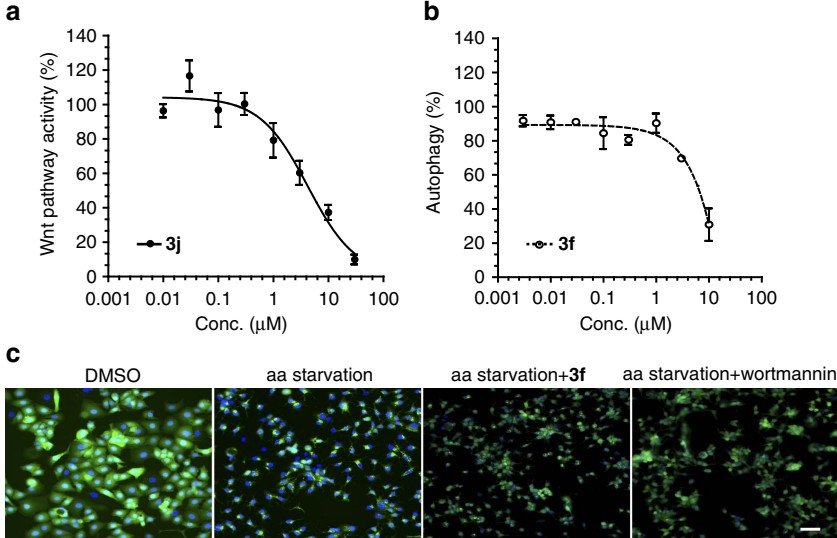

**Figure 8 | Influence of representative compounds on Wnt signalling (3j) and autophagy (3f).** (**a**) Dose-dependent inhibition of the Wnt pathway as determined by means of Wnt reporter gene. HEK293 cells stably transfected with the human Frizzled-1 receptor and a TOPFLASH-driven luciferase reporter gene were treated with different concentrations of **3j** for 6 h. Expression of the firefly luciferase as a reporter gene was the determined by means of luminescence as readout. Nonlinear regression analysis was performed using a four parameter fit. Data are mean values of three independent experiments ($n = 3$) ± s.d. (**b**) Dose–response curve for inhibition of autophagy by **3f**. MCF7-GFP-LC3 cells were deprived of amino acids to induce autophagy and treated with the different concentration of **3f** for 3 h. GFP-LC3 was detected as a measure of autophagosome formation. Nonlinear regression analysis was performed using a four parameter fit. Data are mean values of three independent experiments ($n = 3$) ± s.d. (**c**) MCF7 cells that stably express GFP-LC3 (green) were starved for amino acids (aa) in the presence of the compound **3f** (10 µM) or Wortmannin (3 µM) for 3 h before fixation and staining of the DNA using Hoechst 33342 (blue). Autophagy induction is detected as an accumulation of GFP-LC3 puncta on starvation. Scale bar, 10 µm.

chromatin remodelling complexes to initiate the expression of Wnt target genes. In many epithelial cancers, the Wnt pathway is constitutively active as a result of mutations in different components of the pathway.

We screened the compound collection for modulation of Wnt signalling using a HEK293 reporter cell line that was stably transfected with the human Frizzled-1 receptor and a TOPFLASH-driven luciferase reporter gene[54]. Dose–response analyses were carried out for hit compounds for which cell viability remained >80% with respect to control experiments. To rule out any direct inhibition of firefly luciferase or interference with transcription or translation, hit compounds were assayed for modulation of luciferase in HEK293 cells with constitutive luciferase expression. Pleasingly, the compound collection revealed hits belonging to the quinolone sub-library. For instance compound **3j** (Fig. 8a, $IC_{50} = 4.2 \mu M$) is a structurally novel Wnt pathway inhibitor, which does not interfere with Hh signalling (Supplementary Table 7).

The compound collection was further assessed for modulation of autophagy. Autophagy is a catabolic self-digestive process that is important for maintaining cell homoeostasis through degradation of cellular components, for example, unfolded proteins and damaged organelles like mitochondria, ribosomes and the endoplasmic reticulum, which ensures their regular turnover[55]. Autophagy protects cells under stress conditions, is regarded as a survival mechanism and may provide energy and metabolites to cancer cells that usually are under metabolic stress for the lack of oxygen and nutrients and facing an increased energy demand. Thus, inhibition of autophagy is a promising strategy for targeting tumour cells or sensitizing cancer cells to anti-cancer therapies[56]. The compound collection was screened for autophagy modulation in MCF7 cells that stably express the autophagosome marker LC3 coupled to green fluorescent protein (GFP) to monitor accumulation of GFP-LC3-II on induction of autophagy by amino-acid starvation.

The phenotypic assay revealed that several quinolone and *df*-oxindole derivatives dose-dependently inhibited autophagy with $IC_{50}$ values in the low micromolar range. The quinolone **3f** was the most effective molecule in modulating autophagy with an approximate $IC_{50}$ value of 4.8 µM (Fig. 8b,c).

Notably, quinolone **3f** displayed no activity in the Hh and the Wnt signalling assays, and Hh pathway inhibitors **14q** and **14r** and Wnt signalling inhibitor **3j** did not influence autophagy (Supplementary Table 7).

Finally, we could identify modulators of cell proliferation by means of time-lapse imaging using an IncuCyte ZOOM device, monitoring cell confluency. The assay revealed spirooxindole-bearing compounds as promising class of bioactive compounds affecting proliferation in HeLa cells. Spirooxindole **2d** reduced the proliferation of HeLa cells with an $IC_{50}$ of 15.4 µM (Supplementary Fig. 3a,b; Supplementary Movies 1 and 2), but displayed no activity in the Hh and Wnt signalling assay and in modulation of autophagy. Vice versa, autophagy inhibiting quinolone **3f**, Hh signalling inhibitors **14q** and **14r** and Wnt pathway inhibitor **3j** did not reduce cell proliferation at comparable concentrations and thus are selective inhibitors of the respective pathway/process (Supplementary Table 7).

Although the scaffolds **2**, **3**, **4** and **14** do not hold reactive Michael acceptor functionalities often responsible for covalent binding to cellular targets under mild acidic conditions, nucleophilic addition to olefins might occur. To address this concern, stability of the compounds in the presence of glutathione (GSH) was determined for each scaffold (**2d**, **3f**, **3j**, **4i**, **14q–r**) and after different incubation times (1, 24 and 48 h) in 5 mM GSH in phosphate-buffered saline (PBS; Supplementary Table 8). These compounds representing different scaffolds were stable under these conditions. Only for the autophagy inhibitor **3f** a slight reactivity was detected after incubation for 48 h. However, the mode of action of **3f** cannot be attributed to this reactivity as the treatment time in the autophagy assay is only 3 h. Therefore, the

compounds appear to have a non-covalent and reversible mode of target binding/inhibition.

Thus, the compound collection represented by three different scaffolds generated by ligand-directed gold-catalysed rearrangements of 1,6-enynes successfully delivered structurally diverse small molecules selectively active in four different biological assays monitoring major cellular pathways and processes with relevance to human disease.

## Discussion

Organic synthesis has explored only a limited area of chemical space as represented by a small percentage of scaffolds making up the known synthetic small molecules[57]. A similar lack of structural diversity in screening collections remains a major reason for the limited efficiency in translating small-molecule synthesis into pharmaceutical application. Thus, novel synthetic designs are in high demand, in particular concise catalytic synthesis approaches that can be used to efficiently create scaffold diversity in compound collections that are expected to yield highly useful novel small-molecule candidates for drug and probe discovery research. The ligand-directed synthesis strategy described here is a unified synthetic approach, in which common starting materials are exposed to a common mode of metal catalysis leading to a common type of intermediate, whose molecular fate then was steered to yield three structurally different and diverse scaffolds by tuning the metal catalyst through different ligands. Following this approach and using oxindole-derived 1,6-enynes, we successfully established the reaction conditions, wherein different gold(I) complexes steered the substrates to different reaction pathways thereby generating three distinct scaffold classes. Analysis of a compound collection synthesized by means of this ligand-directed synthesis approach for modulation of major cellular signalling pathways and programs revealed structurally diverse, selective and novel modulators of bioactivity. We believe that the unified scaffold diversity synthesis approach that explores the potential of reactive intermediates, whose molecular fate can be selectively steered towards different structures may also be applicable to other modes of catalysis, for instance in palladium[6,58] and organocatalyzed divergent transformations.

## Methods

**Biological materials.** Dulbecco's Modified Eagle's medium (DMEM), L-glutamine, sodium pyruvate, penicillin/streptomycin, fetal bovine serum (FBS) and fetal calf serum (FCS) were obtained from PAN Biotech, Germany. Chemiluminescent substrate CDP-Star was purchased from Roche, Switzerland. Dual-Luciferase Reporter Assay System was obtained from Promega, USA. pGEN mSmo (Addgene no. 37673) was obtained from Addgene, USA. Anti-acetylated tubulin antibody (Sigma-T6793; 1DB-001-0000868584) and anti-Smo antibody (Abcam-ab38686; 1DB-001-0000338068) were purchased from Sigma, Germany and Abcam, UK, respectively. Alexa Fluor 594-conjugated goat anti-rabbit (A11012; 1DB-001-0000889857) and Alexa Fluor 488-conjugated donkey anti-mouse (A21202; 1DB-001-0000889982) antibodies were purchased from Invitrogen, USA.

**Cell lines.** The murine fibroblast cell line NIH/3T3 was obtained from DSMZ, Germany (DSMZ ACC 59) and was cultured in DMEM (high glucose) supplemented with 10% heat-inactivated FCS, 2 mM L-glutamine and 1 mM sodium pyruvate. Shh Light II cells[59] (NIH/3T3 cells stably transfected with a Gli-responsive firefly luciferase reporter plasmid and a pRL-TK constituitive Renilla luciferase expression vector) were cultured in the same culturing medium as the parental NIH/3T3 cells supplemented with 400 μg ml$^{-1}$ G418 and 150 μg ml$^{-1}$ Zeocin as selecting agents. The murine osteoblasts C3H10T1/2 (ATCC CCL-226) were obtained from ATCC, USA and were cultured in DMEM (high glucose) supplemented with 10% heat-inactivated FCS, 6 mM L-glutamine, 1 mM sodium pyruvate, as well as penicillin and streptomycin. HEK293T cells (ATCC, Middlesex, England) were grown in DMEM containing 10% FBS, 100 U ml$^{-1}$ penicillin and 0.1 mg ml$^{-1}$ streptomycin. All cell lines were maintained at 37 °C and 5% CO$_2$ in humidified atmosphere. All the cell lines were regularly assayed for mycoplasma and were confirmed to be mycoplasma-free.

**Screening to identify modulators of Hh signalling.** For high-throughput screening an osteogenesis assay was performed at the Compound Management and Screen Center (COMAS) in Dortmund, Germany. Briefly, 800 C3H10T1/2 cells were seeded per well in white 384-well plates. After incubation overnight compounds were added to a final concentration of 10 μM using the acoustic nanoliter dispenser ECHO 520 (Labcyte). One hour later, osteogenesis was induced with 1.5 μM purmorphamine. The activity of alkaline phosphatase was measured using the chemiluminescent substrate CDP-Star 96 h after compound addition. One hour after addition of CDP-Star, luminescence was monitored using the Paradigm plate reader (Molecular Devices, USA).

The effect of test compounds on the viability of C3H10T1/2 cells was determined by the CellTiter-Glo Luminescence cell viability assay (Promega). Cells were treated as described for the osteogenesis assay before addition of the CellTiter-Glo reagent and the assay was performed according to manufacturer's protocol. Compounds causing at least a 50% reduction in the osteogenesis assay and retaining cell viability of at least 80% were considered as hits. Dose–response analysis was performed for all hit compounds using a threefold dilution series starting from a concentration of 30 μM. IC$_{50}$ was calculated using the Quattro software suite (Quattro Research GmbH).

**Gli-dependent reporter gene assay.** Shh Light II cells stably expressing a Gli-responsive firefly luciferase reporter plasmid and a pRL-TK vector for constitutive expression of Renilla luciferase were seeded in 96-well plate ($2.5 \times 10^4$ cells per well). After overnight incubation, medium was replaced by low-serum-containing medium (0.5% FCS) supplemented with 1.5 μM purmorphamine. One hour later various concentrations of the compounds or dimethyl sulfoxide (DMSO) as a control were added and cells were further incubated for 48 h. Firefly and Renilla luciferase activity were determined by means of the Dual-Luciferase Reporter Assay System (Promega) according to the manufacturer's protocol using the Infinite M200 plate reader (Tecan, Austria).

**SAG competition assay.** Competition assay was performed using the Gli-dependent reporter gene assay as described above with exception of using 0.1 μM or 1 μM Smo agonist (SAG) to activate the pathway instead of purmorphamine

**Quantitative PCR.** C3H10T1/2 cells were seeded in 24-well plates ($2 \times 10^4$ per well). After overnight incubation, cells were treated with 1.5 μM purmorphamine and the test compounds or DMSO for 48 h. Complementary DNA was prepared using FastLane Cell cDNA Kit (Qiagen) following the manufacturer's instructions. The expression of Hh target genes *Ptch1* and *Gli1* and the reference gene *Gapdh* gene was determined by means of quantitative PCR using the QuantiFast SYBR Green PCR Kit (Qiagen) and iQ 5 Real-Time PCR Detection System (Bio-rad) and the following oligonucleotides: *Ptch1*: 5′-CTCTGGAGCAGATTTCCAAGG-3′ and 5′-TGCCGCAGTTCTTTTGAATG-3′, *Gli1*: 5′-GGAAGTCCTATTCACGCCT TGA-3′ and 5′-CAACCTTCTTGCTCACACATGTAAG-3′, *Gapdh*: 5′-AGCCTCG TCCCG TAGACAAAAT-3′ and 5′-CCGTGAGTGGAGTCATACTGGA-3′[60]. The expression levels of *Ptch1* and *Gli1* were determined using the $2 - \Delta\Delta Ct$ method[61].

**Smo binding assay using fluorescence microscopy.** A total of $1.5 \times 10^4$ HEK293T cells were seeded on poly-D-lysine-coated coverslips placed in a 24-well plate. After overnight incubation, cells were transfected with the Smo expressing plasmid pGEN-mSmo (Addgene no. 37673) using Fugene HD (Promega) according to the manufacturer's protocol. After 48 h incubation at 37 °C, cells were washed twice with PBS and fixed with 3% paraformaldehyde for 10 min at room temperature and subsequent permeabilization with 0.2% sodium azide in 1× PBS for 5 min at room temperature. The cells were washed once with PBS and incubated further in fresh DMEM medium containing 0.5% FBS, 5 nM BODIPY-cyclopamine and various concentrations of the test compounds or DMSO as a control. One hour later cells were washed twice with PBS and stained with 1 μg ml$^{-1}$ 4′,6-diamidino-2-phenylindole for 10 min and were mounted on glass slides using Aqua Poly/mount (Polysciences Inc). Images were acquired on an Axiovert Observer Z1 microscope (Carl Zeiss, Germany) using a Plan-Apochromat × 63/1.40 Oil DIC M27 objective.

**Smo binding assay using flow cytometry.** A total of $3 \times 10^5$ HEK293T cells were seeded per well in six-well plates. After incubation overnight, cells were transfected with the Smo expressing plasmid pGEN-mSmo as described above. Forty-eight hours later medium was replaced by DMEM containing 0.5% FBS, 5 nM BODIPY-cyclopamine and various concentrations of the test compounds or DMSO as a control. Following incubation for 5 h cells were washed once with PBS, detached using trypsin/EDTA (0.05/0.02% in PBS), and collected by centrifugation at 250*g* for 5 min at room temperature. Cells were washed twice and then suspended in ice-cold PBS. Cell suspensions were subjected to flow cytometry analysis employing the BD LSR II Flow Cytometer (laser line: 488 nm, emission filter: 530/30) to detect the presence of BODIPY. Data analysis was performed using the FlowJo software, version 7.6.5 (Tree Star Inc., USA).

**Ciliary localization of Smo.** A total of $3 \times 10^4$ NIH/3T3 cells were seeded per well in 24-well plates containing coverslips and cultured overnight. Cells were further incubated for 12 h in DMEM containing 0.5% FCS to induce ciliation. Cells were then treated with 1.5 μM purmorphamine and various concentrations of the test compounds or DMSO as a control. Twelve hours later cells were washed with PBS followed by fixation in 4% ice-cold paraformaldehyde for 10 min at room temperature. Permeabilization and blocking of non-specific binding was performed with a solution containing 0.1% Triton X-100 and 1% heat-inactivated horse serum in PBS for 30 min at room temperature. Samples were then incubated with mouse anti-N-acetylated tubulin antibody (dilution 1:5,000) and rabbit anti-Smo antibody (dilution 1:200) overnight at 4 °C followed by washing and subsequent incubation with Alexa Fluor 594-conjugated goat anti-rabbit and Alexa Fluor 488-conjugated donkey anti-mouse antibodies (Invitrogen; 1:500 dilutions) and 4′,6-diamidino-2-phenylindole (0.1 μg ml$^{-1}$) for 45 min at room temperature. Coverslips were washed and mounted using Aqua Poly/mount (Polysciences. Inc USA). Images were acquired in a Axiovert Observer microscope Z1 (Carl Zeiss, Germany) using a Plan-Apochromat × 63/1.40 Oil DIC M27 objective.

**Screening to identify inhibitors of Wnt signalling.** The screening was performed at COMAS in Dortmund, Germany. Wnt activity was analysed in HEK293 human embryonic kidney cells stably transfected with the human Frizzled-1 receptor and a luciferase reporter gene under the control of the TOPFLASH motif[54]. Cells were activated by addition of Wnt 3a-conditioned medium that had been collected from L cells overexpressing the Wnt 3a protein[13]. Control cells were treated with conditioned medium from L cells. Three thousand HEK293-Fz1-TOPFLASH cells were seeded per well of a 384-well plate and incubated with the compounds for 6 h. XAV939 was used as a control for Wnt pathway inhibition. OneGlo Reagent (Promega Inc.) was added and the luminescence signal was recorded with a Spectramax Paradigm reader (Molecular Devices). As toxic compounds also lead to a decreased luciferase signal, cell viability was assessed in parallel to the Wnt reporter gene assay using the CellTiter-Glo reagent (Promega). Briefly, 1,000 HEK293-Fz1-TOPFLASH cells were seeded per well in a 384-well plate. After a treatment with the compounds for 6 h, the luminescence signal was measured after addition of the CellTiter-Glo reagent. Dose–response analyses were carried out using eight threefold dilution steps starting from 30 μM. Compounds were dispensed using an Echo520 acoustic dispenser (Labcyte Inc.). IC$_{50}$ values were calculated with Quattro/Workflow (Quattro research GmbH). All measurements were performed in triplicate. To identify compounds that interfere with transcription or translation or directly inhibit luciferase, a HEK293 cell line with constitutive firefly luciferase expression was used. Three thousand cells were seeded per well in a 384-well plate and incubated with the compounds for 6 h. OneGlo Reagent (Promega) was added and the luminescence signal was recorded with a Spectramax Paradigm reader (Molecular Devices). Control cells were treated with 10 μM cycloheximide. Dose–response analyses were carried out as described above.

**Screening to identify autophagy modulators.** Screening for autophagy modulators was performed at COMAS in Dortmund, Germany. Briefly, MCF7-GFP-LC3 (4,000 cells per well) cells were seeded in 384-well plates (Greiner). After incubation overnight cells were washed thrice with PBS using the plate washer ELX405 (Biotek) followed by addition of 10 μM of the compounds using the Echo dispenser (Labcyte) along with Earl's Balanced Salt Solution to induced autophagy and chloroquine (50 μM). Cells were incubated for 3 h before fixation using 4.6% formaldehyde in PBS and staining of the DNA using Hoechst 33342 for 20 min at room temperature. Fixed cells were washed thrice with PBS using the plate washer ELX405 (Biotek). For visualization, four pictures per well were acquired on an ImageXpress Micro XL (Molecular Devices) at × 20 magnification and analysed with the granularity setting of the MetaXpress Software (Molecular Devices). Wortmannin was used as a control for autophagy inhibition. Dose–response analysis was carried out using eight doses of a threefold dilution series starting at 10 μM. IC$_{50}$ values were calculated using the Quattro Workflow software (Quattro Research GmbH).

**Time-lapse proliferation study in HeLa cells.** HeLa cells (3,000 cells per well) were seeded in 96-well clear bottom plates and incubated overnight at 37 °C. On the following day, the medium was replaced by fresh medium containing various concentrations of test compounds and DMSO as a control. Cells were then subjected to live-cell imaging over 96 h under standard incubation conditions using an IncuCyte ZOOM imaging system (Essen BioScience). Images were acquired every 30 min. Image-based analysis of cell confluence was carried out using the IncuCyte2015A software. All the experiments were performed in triplicates.

**GSH reactivity assay.** 30 μM of the compounds were incubated with PBS or 5 mM GSH for different period of time at 37 °C. Samples were then analysed by high-performance liquid chromatography–mass spectrometry (HPLC: Ultimate 3000, MS: VELOS Pro- Thermo Fisher Scientific, Waltham, MA, USA). Peak areas for the respective molecular weight of the compound incubated with PBS or GSH was employed for comparison. The peak areas were normalized to the respective peak areas in PBS to obtain the per cent abundance of the compounds in GSH.

**Chemical synthesis.** Compounds were synthesized according to the procedures specified in Supplementary Methods. X-ray crystallographic data and images are reported in Supplementary Tables 3–6. For $^1$H and $^{13}$C nuclear magnetic resonance spectra of compounds see the section of nuclear magnetic resonance spectra in Supplementary Figs 4–134.

**Data availability.** The X-ray crystallographic coordinates for **2a**, **3a**, **4a** and **14a** can be obtained free of charge from The Cambridge Crystallographic Data Centre via www.ccdc.cam.ac.uk/data_request/cif under deposition numbers CCDC 1448677, 1448646, 1448652 and 1448645, respectively. All other data supporting the findings of this study are available within the paper and its Supplementary Information Files.

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

## Acknowledgements

The research was supported by the Max Planck Gesellschaft and the European Research Council under the European Union's Seventh Framework Programme (FP7/2007–2013)/ ERC Grant agreement no. 268309. The Compound Management and Screening Center (COMAS, Dortmund) is acknowledged for carrying out the screening and data analysis. Y.-C.L. like to thank Ministry of Education (Taiwan) for the MOE Technologies Incubation Scholarship.

## Author contributions

Y.-C.L. performed chemical experiments. S.P. performed biological experiments. C.G. and C.S. determined crystal structure analyses. S.Z., K.K. and H.W. supervised the research. Y.-C.L., S.P., S.Z., K.K. and H.W. designed experiments and wrote the manuscript.

## Additional information

**Competing financial interests:** The authors declare no competing financial interests.

**Reprints and permission** information is available online at http://npg.nature.com/ reprintsandpermissions/

