## [Peer Review File · Nature Communications]

Reviewers' comments:

Reviewer #1 (Remarks to the Author):

The authors describe a divergent synthesis of spirooxindoles, quinolones or di-oxindoles from oxindole-derived 1,6-enynes. The chemoselectivity is achieved by changing different ligands. Moreover, the synthesized compound collection are treated in cell-based assays and several molecules are found to have biological activities. The work itself is very interesting and the obtained three scaffolds are structurally novel, but the manuscript suffers from several issues that need to be addressed before this work can be accepted for publication. I therefore recommend publication pending consideration of the following points that are offered in a constructive manner.

1) Fig. 1 describes the strategy for the diverse synthesis. This can not be a strategy because I don't think the authors have known the results of the reactions at the beginning of the research. And also I am confused if this is a ligand-directed synthesis, do the substituents also determine the reaction pathways? I suggest that the significant usefulness of the spiro-indole scaffolds may be a good topic to replace this part.

2) Other ligand-directed divergent synthesis using gold catalysts have been reported, for example, *J. Am. Chem. Soc.*, 2015, 137 (25), 8131-8137; *Chem. Sci.*, 2016, DOI: 10.1039/C6SC00058D, *Chem. Eur. J.* 2015, 21, 7675-7681; *J. Am. Chem. Soc.*, 2016, 138 (16), 5218-5221 and other related papers. Those works should be cited and briefly discussed in somewhere of the MS.

3) As a result, Page 22, the last sentence: 'We believe...divergent transformations' is overstated and need to be deleted.

4) Products 14 are formed as single anti-diastereoisomers. It is not explained in the MS. The stereochemistry in intermediates 15-19 (Fig. 5) should be assigned. Can gold center coordinates with the carbonyl group, or it has to stand at the opposite site of carbonyl group due to steric hindrance?

5) Stereochemistry in Fig. 3 should also be carefully considered.

6) 5 mol% of gold catalyst is used in the reaction. Transition metal residual may contaminate the sample. Would it be a problem to the assays? Could the loading of catalyst be reduced?

7) Some of the products are not very pure according to NMR. For example, 3i.

Overall, the authors have found a very interesting divergent synthesis of three types of heterocycles from oxindole-derived 1,6-enynes. The cell-based assays showed some of those small libraries may have promising biological activities. I look forward to reading more from this group in the near future.

Reviewer #2 (Remarks to the Author):

In this manuscript, Waldmann and co-workers described gold catalyzed cycloisomerization of oxindole-derived enynes, which afforded three types of structurally distinct molecules by altering the ligand coordinated to gold complex and the products displayed the potential of being selective modulators of the Hedgehog- and Wnt signaling pathways, autophagy and cellular proliferation. Although the scaffolds of products are distinct, the gold(I) catalyzed enyne cycloisomerizations chemistry was not in convinced originality. The switchable chemistry achieved via tuning electronic and steric properties of ligands has been reported as well. And the efficiency was feeble, in which many examples are in low yields. All in all, the work is not qualify for Nat Commun.

1. Methanol was chosen as nucleophile, how about other nucleophiles that often applied in enyne cycloisomerization such as water, acid and other alcohol?
2. Which is the reason for low yields, incomplete consumption of SM or formation of byproducts?
3. Some important references have been missing. The first example of gold(I) catalyzed enyne cycloisomerization: *Angew. Chem., Int. Ed.* 2004, 43, 2402-2406.
4. Ligand controlled regiodivergent gold(I) catalyzed reactions: (a) *J. Am. Chem. Soc.* 2005, 127, 10500. (b) *Organometallics* 2006, 25, 2237. (c) *J. Am. Chem. Soc.* 2008, 130, 6940. (d) *J. Am. Chem. Soc.* 2009, 131, 6348. (e) *J. Am. Chem. Soc.* 2009, 131, 13020. (f) *Angew. Chem., Int. Ed.* 2010, 49, 2542. (g) *Org. Lett.* 2011, 13, 5580. (h) *Chem.-Eur. J.* 2012, 18, 6811. (i) *Chem.-Eur. J.* 2012, 18, 15113. (j) *J. Am. Chem. Soc.* 2014, 136, 8887. (k) *J. Am. Chem. Soc.* 2014, 136, 13146. (l) *J. Am. Chem. Soc.* 2015, 137, 6350. (m) *J. Am. Chem. Soc.* 2015, 137, 8131.
5. Typos
 - 1) Line 100: "biraryl"
 - 2) Line 102: "equally"

Reviewer #3 (Remarks to the Author):

The authors first describe the synthesis of small libraries of compounds based upon several different yet related scaffolds from a common intermediate. This appears to this reviewer as an elegant and novel chemical approach, however, I am not eminently qualified to comment on this particular aspect of the manuscript.

The authors then go on to attempt to demonstrate broad biological activity of the libraries that they constructed, using several different assays (Hedgehog inhibition, Wnt signaling inhibition, inhibition of autophagy and cellular proliferation). One aspect of these libraries that is somewhat concerning in regards to the various scaffolds that were constructed (2, 3, 4 and 14) is that they are all in essence weak electrophiles that could potentially react with biologically relevant nucleophiles (e.g. free sulfhydryl groups), thereby potentially compromising specificity/selectivity.

The authors primarily describe the screening for Hedgehog inhibition using a variety of formats. Inhibition of purmorphamine induced osteogenesis in C3H10T1/2 mesenchymal progenitors demonstrated significant and very similar inhibitory activity for two closely related analogs (14 q and r). Given the structural similarity of the 14 series and related nature of the 4 series compounds, and the overall relatedness of the compounds, the authors should provide the primary screening results for all compounds assayed to further demonstrate the unique nature of their library and the specificity both inter and intra-target (this could be provided in a supplementary table). It appears in all of the Hedgehog assay formats

(details of the other assays are not provided) that the compounds were incubated overnight before testing. Is this necessary because a relatively slow "electrophilic"

reaction is required for significant inhibitory activity? Do the compounds display similar IC₅₀ values if incubated only for 30min prior to initiation of the assay?

Further in that regard, is the concentration dependent displacement of BODIPY-cyclopamine reversible or irreversible (this could be tested by a washout experiment).

In regards to the additional biological assay evaluation, the results presented are very cursory at best. Testing in HEK293 cells (not generally transfected with human Fzd-1) is standard for Wnt inhibition, as HEK 293 cells readily respond to

various mechanisms of Wnt stimulation (i.e. LiCl, GSK3 inhibitor or Wnt3a) and do not require Fzd-1 transfection for this purpose. However, these cells need to be

stimulated to demonstrate significant Topflash activity. No assay details are provided i.e. incubation time, method of stimulation, control compound(s). The only data provided states that 3j has an IC₅₀=4.2 μM. Again providing the assay results

for minimally all structurally related 3 series compounds and preferentially all compounds assayed would be very beneficial. It is stated that 3j is specific for Wnt inhibition as it does not inhibit Hedgehog signaling? This is very difficult to assess

without presentation of the counter-screening results.

The phenotypic autophagy assay data presented (Fig 8) is not very convincing, although an IC₅₀ value of 4.8 μM for compound 3f is stated. How was this assay quantified? A dose response should

be provided as in Fig 8, compound 3f was tested at 10 μM (i.e. 2X the IC₅₀) yet to this reviewer the inhibition of autophagy at this concentration is marginal at best. Again a positive control compound should be included for comparison.

Finally, cell proliferation in HeLa cells was tested. The most potent compounds were from series 2 (2d IC₅₀=15.4 μM). Although, judging from supplementary Fig 2, many of the series 2 compounds showed similar levels of activity.

Furthermore, it is somewhat surprising to this reviewer that compounds that inhibit Wnt and HH signaling in cell based assays at low μM levels did not effect HeLa cell proliferation.

Significant additional biological data should be provided in

order to justify the authors' statement for "selectively active in four biological assays monitoring major cellular pathways and processes with relevance to human disease".

Reviewer #1 (Remarks to the Author):

1) *Fig. 1 describes the strategy for the diverse synthesis. This cannot be a strategy because I don't think the authors have known the results of the reactions at the beginning of the research. And also I am confused if this is a ligand-directed synthesis; do the substitutes also determine the reaction pathways? I suggest that the significant usefulness of the spiro-indole scaffolds may be a good topic to replace this part.*

Authors' response: Thanks for the comment which is well taken and the wording has been changed accordingly in the text as well as in the caption of the Fig. 1.

2) *Other ligand-directed divergent synthesis using gold catalysts have been reported, for example, J. Am. Chem. Soc., 2015, 137 (25), 8131-8137; Chem. Sci., 2016, DOI: 10.1039/C6SC00058D, Chem. Eur. J. 2015, 21, 7675-7681; J. Am. Chem. Soc., 2016, 138 (16), 5218-5221 and other related papers. Those works should be cited and briefly discussed in somewhere of the MS.*

Authors' response: The references are now cited in the revised manuscript (19-22) and also mentioned in the text in the first paragraph of the results and discussion.

3) *As a result, Page 22, the last sentence: 'We believe...divergent transformations' is overstated and need to be deleted.*

Authors' response: The sentence cites references for related work and ideas already realized in palladium chemistry, thus it is not only an expectation, but draws from work pointing towards a new emerging principle. Also the scope of divergent reaction pathways using common mode of organocatalysis is definitely possible and a potential area of research in organic synthesis. Thus, we think that we did not overstate the case here.

4) *Products 14 are formed as single anti-diastereoisomers. It is not explained in the MS. The stereochemistry in intermediates 15-19 (Fig. 5) should be assigned. Can gold center coordinates with the carbonyl group, or it has to stand at the opposite site of carbonyl group due to steric hindrance?*

Authors' response: Thanks for the comments. The information regarding 14 formed as single diastereoisomers has been added both in the main text and in the supporting information.

The carbons for which stereochemistry can be traced back from the products to the intermediates (15-19) have been shown in fig. 5. For the spiro-carbon, we cannot depict a stereochemistry in 15-19 as this information is lost in the subsequent rearrangement and therefore assigning a stereochemistry at this center may be misleading. We trust the reviewer understands that we would not go too far here.

We don't anticipate that the gold center will coordinate to the carbonyl function because the two rings on a spiro-center are orthogonally placed which keeps the carbonyl moiety quite at a distance from the gold center.

5) 5 mol% of gold catalyst is used in the reaction. Transition metal residual may contaminate the sample. Would it be a problem to the assays? Could the loading of catalyst be reduced?

Authors' response: Supporting table 1 now has been added to provide further details about the reaction condition optimization for diverse scaffold synthesis, side product formed or incomplete conversions etc. Similar optimization table (SI Table 2) for 14a is also included in the supporting information. In the synthesis of compound class **2**, **4**, **14**, reduction of catalyst loading to 3 mol% afforded slightly lower yield (entry 10 and 28, SI table 1; entry 13, SI table 2). However, in the compound class **3**, the yield drastically decreased from 67% (5 mol% catalyst) to 27% (3 mol% catalyst) (entry 7, SI table 1).

In case of contamination by transition metal catalyst, we could have observed many false positives. Fortunately, that is not the case. On the one hand, the active molecules were purified by HPLC in order to have high purity before their activity in assays were measured, on the other hand, gold complexes in trace amounts are not known to cause cellular toxicity. Importantly, if we had consistent gold contamination leading to cellular effects, we would hardly have found selective and different bioactivity. Therefore, we are confident of the *real* activity displayed by these molecules. We have applied these criteria and demands for our synthesis products since quite some time on compound classes emerging from different syntheses, and we found that these criteria are state of the art and sufficient to assure reliable biological results.

7) Some of the products are not very pure according to NMR. For example, 3i.

Authors' response: We have updated the following ¹H spectra, **S8j**, **3c** (¹H and ¹³C), **3i**, **3j**, **13a**, **13t**, and **14b**, in the revised supporting information.

Reviewer #2 :

1. Methanol was chosen as nucleophile, how about other nucleophiles that often applied in enyne cycloisomerization such as water, acid and other alcohol?

Authors' response: For the nucleophile screening, we selected water, acetic acid, and indole to perform the *df*-oxindole formation reaction. In case of water as nucleophile, the desired product **4OH** was formed in low yield (23%) along with mixture of epimeric quinolones in 45% yield (syn: anti = 2:1), plausibly due to the heterogeneous nature of the reaction in 1,2-dichloroethane in the presence of water (entry 24, SI table 1). When acetic acid was applied as nucleophile, the desired product was obtained again as diastereomeric mixture in 56% yield (**4OAc** : **epi-4OAc** = 4:3). The relatively more stable carbocation in this case could steer the reaction, which might be due to the stabilization of carbocationic intermediate, and epimeric quinolones were also observed in 20% yield (syn: anti = 5:3) (entry 25, SI table 1). Indole was selected as a representative carbon nucleophile; however the poor nucleophilicity afforded no addition product *albeit* quinolone formation (37% yield) was observed along with Meyer-Shuster product in 23% yield (entry 26, SI table 1).

Under the optimal condition, ethanol as nucleophile successfully provided the desired product **4n** in 76% yield and has now been added to the Figure 2 as new result.

2. Which is the reason for low yields, incomplete consumption of SM or formation of byproducts?

Authors' response: We have characterized the side products in each optimized reaction condition. In the case of quinolone (**3**) formation, the major side product is the epimeric quinolone **epi-3a** (20% yield), which was generated from the (*Z*)-1,6-enyne (entry 6, SI table1). In case of formation of spirooxindole (**2**) formation, the unreactive (*Z*)-1,6-enyne was isolated in 10% yield along with some unidentifiable compounds (entry 9, SI table1). When MeOH was used as nucleophile, besides 73% yield of *df*-oxindoles (**4**), fumaric ketone was also formed via gold catalyzed Meyer-Shuster rearrangement in 18% yield (entry 24, SI table1). All these details are now summarized in Supporting table 1.

3. Some important references have been missing. The first example of gold(I) catalyzed enyne cycloisomerization: *Angew. Chem., Int. Ed.* 2004, 43, 2402-2406. Ligand controlled regiodivergent gold(I) catalyzed reactions: (a) *J. Am. Chem. Soc.* 2005, 127, 10500. (b) *Organometallics* 2006, 25, 2237. (c) *J. Am. Chem. Soc.* 2008, 130, 6940. (d) *J. Am. Chem. Soc.* 2009, 131, 6348. (e) *J. Am. Chem. Soc.* 2009, 131, 13020. (f) *Angew. Chem., Int. Ed.* 2010, 49, 2542. (g) *Org. Lett.* 2011, 13, 5580. (h) *Chem.-Eur. J.* 2012, 18, 6811. (i) *Chem.-Eur. J.* 2012, 18, 15113. (j) *J. Am. Chem. Soc.* 2014, 136, 8887. (k) *J. Am. Chem. Soc.* 2014, 136, 13146. (l) *J. Am. Chem. Soc.* 2015, 137, 6350. (m) *J. Am. Chem. Soc.* 2015, 137, 8131.

Authors' response: New references as suggested by 1st and 3rd reviewer have been added in the revised manuscript.

5. Typos 1) Line 100: " biraryl"2) Line 102: "equally"

Authors' response: The typos are corrected now.

Reviewer #3 :

1. One aspect of these libraries that is a somewhat concerning in regards to the various scaffolds that were constructed (2, 3, 4 and 14) is that they are all in essence weak electrophiles that could potentially react with biologically relevant nucleophiles (e.g. free sulfhydryl groups), thereby potentially compromising specificity/selectivity.

Authors' response: This is a valid concern. Although the scaffolds do not hold reactive Michael acceptor functionalities (tetrasubstituted olefins in **3**, **4** and **14**) under mild acidic conditions, nucleophilic addition to olefins might happen. To address this concern of the learned reviewer, reactivity of the compounds with glutathione (GSH) was determined for each scaffold (**2**, **3**, **4** and **14**) and after different incubation times (1, 24 and 48 h). Only for the autophagy inhibitor **3f** a slight reactivity was detected after incubation for 48 h. However, the mode of action of **3f** cannot be attributed to this reactivity as the treatment time in the autophagy assay is only 3 h. Therefore, the studied compounds seem to have a non-covalent mode of target binding /inhibition. The data has been added to the manuscript (see Supplementary Table 8).

2. Given the structural similarity of the 14 series and related nature of the 4 series compounds, and the overall relatedness of the compounds, the authors should provide the primary screening results for all compounds assayed to further demonstrate the unique nature of their library and the specificity both inter and intra-target (this could be provided in a supplementary table).

Response: We have included the Supplementary Table 7 that displays the results of the primary screening.

3. It appears in all of the Hedgehog assay formats (details of the other assays are not provided) that the compounds were incubated overnight before testing. Is this necessary because a relatively slow "electrophilic" reaction is required for significant inhibitory activity? Do the compounds display similar IC50 values if incubated only for 30min prior to initiation of the assay?

Authors' response: To obtain a measurable response of Hedgehog signalling activation, cells have to be incubated with the inducer purmorphamine for 48 or 96 hours depending on the cell line. For this reason, test compounds were applied for 48 or 96 h as well together with purmorphamine. There is no pre-incubation with the compounds overnight. Instead test compounds are added one hour prior to induction with purmorphamine (see section *Biological Materials and Methods*). Weak electrophilic reactivity we have excluded (see above).

4. Further in that regard, is the concentration dependent displacement of BODIPY-cyclopamine reversible or irreversible (this could be tested by a washout experiment).

Authors' response: To address the mode of inhibition (reversible or irreversible) we performed a washout experiments with compounds **14q** and **14r**. After incubation with the compounds for 30 min, compounds were removed by extensive washing and the Hedgehog pathway activity was assessed using the Gli reporter gene assay. The activity of both compounds was substantially lower after the washout and no IC₅₀ values could be determined. We have included the data in the manuscript (see Supplementary Figure 2a).

In addition, we included a washout step in the BODIPY-cyclopamine displacement assay. Washout of **14q** and **14r** after incubation of cells for one hour abolished the displacement of

BODIPY-cyclopamine by the compounds. Thus, these small molecules show a reversible mode of action. These results of included in the Supplementary Figure 2b.

5. In regards to the additional biological assay evaluation, the results presented are very cursory at best. Testing in HEK293 cells (not generally transfected with human Fzd-1) is standard for Wnt inhibition, as HEK 293 cells readily respond to various mechanisms of Wnt stimulation (i.e. LiCl, GSK3 inhibitor or Wnt3a) an do not require Fzd-1 transfection for this purpose. However, these cells need to be stimulated to demonstrate significant Topflash activity.

Authors' response: We agree with the reviewer that the parental cell line HEK293 already responds to Wnt3a stimulation. However, robust assays are needed for screening. For this reason, the Wnt reporter assay was carried out using a stably transfected HEK293 cell line which contains the Topflash motif. This cell line contains additional copies of the Frizzled receptor which additionally enhances the Wnt pathway activity. The suitability of this dual stable transfected cell lines was demonstrated in several studies (see reference 43 in the manuscript).

No assay details are provided i.e. incubation time, method of stimulation, control compound(s). The only data provided states that 3j has an $IC_{50}=4.2 \mu M$. Again providing the assay results for minimally all structurally related 3 series compounds and preferentially all compounds assayed would be very beneficial. It is stated that 3j is specific for Wnt inhibition as it does not inhibit Hedgehog signaling? This is very difficult to assess without presentation of the counter-screening results.

Authors' response: The assay description is now included in the updated manuscript. The data from the Wnt screening is shown in Supplementary Table 7 to demonstrate the specificity to Wnt inhibition. Additionally, the dose-response curve for **3j** is now included in the updated manuscript and the employed control compound (XAV939) is mentioned in the method section. Moreover, the detected Wnt inhibition depends on the different substituents on series **3** compound. We observe an inhibition of Hedgehog signaling along with Wnt pathway inhibition only for compound **3i**.

6. The phenotypic autophagy assay data presented (Fig 8) is not very convincing, although an IC_{50} value of $4.8 \mu M$ for compound 3f is stated. How was this assay quantified? A dose response should be provided as in Fig 8, compound 3f was tested at $10 \mu M$ (i.e. 2X the IC_{50}) yet to this reviewer the inhibition of autophagy at this concentration is marginal at best. Again a positive control compound should be included for comparison.

Authors' response: The assay description and quantification is now included in the manuscript. A dose-response curve for **3f** is now included in the Figure 8b. As depicted in this figure, autophagy is not completely inhibited at the highest tested concentration of $10 \mu M$ (the residual activity is ca. 30 %). Thus, the IC_{50} value for inhibition of autophagy by **3f** is estimated to be 4.8

μM . We have adjusted the wording in the manuscript and added information on the employed positive control wortmannin as well.

Finally, cell proliferation in HeLa cells was tested. The most potent compounds were from series 2 (2d $\text{IC}_{50}=15.4\mu\text{M}$). Although, judging from supplementary Fig 2, many of the series 2 compounds showed similar levels of activity.

Authors' response: We have included the dose-response curves for compounds **2i**, **2j** and **2d** in Supplementary Figure 3b showing **2d** to be the most potent inhibitor of cell proliferation among the assayed small molecules.

7. Furthermore, it is somewhat surprising to this reviewer that compounds that inhibit Wnt and HH signaling in cell based assays at low μM levels did not effect HeLa cell proliferation.

Authors' response: We do not expect inhibitors of Hedgehog and Wnt signalling to influence proliferation of HeLa cells. These inhibitors should rather show effects only in Wnt- or Hedgehog-stimulated cells.

Significant additional biological data should be provided in order to justify the authors' statement for "selectively active in four biological assays monitoring major cellular pathways and processes with relevance to human disease".

Authors' response: As suggested by the reviewer we have included a table with the results of all primary screenings (see Supplementary Table 7) and various data, as detailed above. The table nicely demonstrates the selective mode of action of the compounds in regard to the four studied cellular processes.

REVIEWERS' COMMENTS:

Reviewer #3 (Remarks to the Author):

The authors have responded adequately to my comments/concerns regarding the biological assay data and methods utilized and I believe the manuscript should be acceptable for publication.